# Are Professionals Rationals? How Organizations and Households Make E-Car Investments

**Ingo Kastner [1],\*, Annalena Becker [1], Sebastian Bobeth [2] and Ellen Matthies [1]**

1 Department of Environmental Psychology, Institute of Psychology,
Otto-von-Guericke-University Magdeburg, Universitätsplatz 2, 29106 Magdeburg, Germany;
annalena.becker@ovgu.de (A.B.); ellen.matthies@ovgu.de (E.M.)

2 Erich Fromm Study Center, International Psychoanalytic University, Stromstr. 1, 10555 Berlin, Germany;
sebastian.bobeth@ipu-berlin.de

\* Correspondence: ingokastner@yahoo.de

**Abstract:** This study attempts to identify the main drivers for e-car investments in households and organizations. We questioned 227 decision makers in households currently considering car purchases, and 101 decision makers in small businesses. The businesses were private care services, because their driving profiles widely fit the capabilities of modern e-cars. The main investment drivers were compared in an integrated action model involving elements of the theory of planned behavior and the norm-activation model, i.e., investment intentions, attitudes, personal (ecological) and social norms, and perceived behavioral control. For each group, different models were calculated in order to investigate the relevance of different types of social norms within the decision process, i.e., injunctive or descriptive norms. As expected, the household and organizational decisions were found to be based on different key factors: the decision makers in households mostly considered personal and descriptive social norms; the organizational decisions were mostly grounded in attitudes and injunctive social norms concerning staff expectations. The results suggest the need for tailored policy measures for each target group.

**Keywords:** mobility investment decisions; e-cars; households; organizations; social norms

## 1. Introduction

Politicians and scientists widely agree that the human way of living and consuming has to change fundamentally if the natural environment is to be preserved for future generations [1]. One key element to reach this goal lies in the energy system. Still, worldwide energy consumption and production is mainly based on fossil fuels, which cause substantial emissions and thus contribute to climate change and other environmental problems [2,3]. In the last few years, several actions have been taken in order to make energy production and consumption more sustainable. Positive examples can especially be found in electricity production, where several countries have reached high shares of renewable energies [3]. Germany is one positive example where the transition in the electricity sector has made significant progress [4].

However, there are barely any developments in other sectors. Especially in the mobility sector, worldwide emissions continue to increase [1]. One key approach in stopping these developments is to replace fossil-fuel cars (f-cars) with those using other engine types. Consequently, several countries have defined phase-out dates for f-cars [5]. These measures are encouraging but insufficient, given the rapid progression of climate change. Most bans will not take effect for decades if they target only new registrations. Thus, additional measures focusing on voluntary actions (e.g., incentive strategies) are necessary to support policy aims. Such measures could also accelerate the phase-out in countries where f-car bans are not yet planned.

The basis for effective policy measures (or interventions) lies in the understanding of the actors' motives. A new behavior can only be appropriately implemented if its drivers

and barriers (including competitive behaviors) are identified (e.g., [6,7]). The differences between the target groups should be considered as well [7,8], as most cars are purchased by either households or organizations. The consumption patterns of both groups are slightly different: while organizations generally prefer new cars, households tend to buy pre-owned vehicles, probably because they are less expensive (see e.g., [9,10] for Germany and the United Kingdom). Nevertheless, there is barley any market for pre-owned cars using alternative engines systems, given that their distribution rate is still low. Consequently, organizations might be the more important target group for alternative car types (e.g., e-cars), as most of the purchased cars are new.

Research concerning mobility-related investments goes in another direction. In the last few years, a large number of studies have been conducted investigating the drivers of mobility-related investments (see e.g., [11,12] for overviews). However, these investigations mostly focus on household decisions, and there is limited research in the organizational sector. Additionally, there is a remarkable imbalance concerning the methods used for these investigations: most household studies involve large samples and quantitative methods (e.g., questionnaires); they also mostly refer to well-established action models depicting the decision drivers (e.g., [13–20]). Organizational research is mostly based on qualitative methods (i.e., interviews) using small samples (e.g., [21–23]), while quantitative analyses and model analyses are scarce (see [24] for an exception). One reason for this imbalance might be that the number of people making mobility investments in organizations is generally smaller compared to household decision makers. Acquiring participants—or even large samples—for quantitative analyses is thus much more complicated.

Taken together, there is at present an increasing amount of research concerning the drivers of mobility investments in households, while there are limited data for organizations. In other words, there is no way to know yet whether both groups base their decision on similar or different factors, and whether policy measures need to be tailored for each group. The scope of this paper is to compare the mobility investment decisions in each group, and to identify the key factors for policy designs. For this, we will draw on well-established action models. These allow us to depict decision factors and to identify the most important ones. In the following, we will first introduce a few models that have already been used in the context of mobility investments. We will also present the available empirical studies referring to these models, and we will discuss the value of integrative approaches. Then, we will present data from two studies on e-car investments in households and organizations. The theoretical and practical implications of our findings will be discussed at the end.

## 2. Theory

Quite a number of different action models are used to explain mobility actions. Most empirical analyses have focused on the everyday mobility patterns of household members (e.g., [6,25] for overviews; see [26] for travel mode choice). However, in the last few years, there has also been an increasing number of studies focusing on mobility investments (e.g., [13–20,24,27,28]). Still, most of them were conducted in households. In the following, we will present some well-established action models that have already been used to depict mobility behavior. The differences between the theoretical approaches will be discussed.

### 2.1. Theory of Planned Behavior

A well-known approach is the theory of planned behavior (TPB), which was introduced by Ajzen (e.g., [29,30]). In this model, one's behavior is understood to be mainly affected by a behavior intention (see Figure 1). This intention is influenced by the attitude towards the behavior (i.e., the expected positive behavioral consequences), subjective norms (i.e., the expected social approval by significant others), and perceived behavioral control (PBC; i.e., the perceived ability to perform the behavior). In some cases, PBC was also found to directly influence behavior.

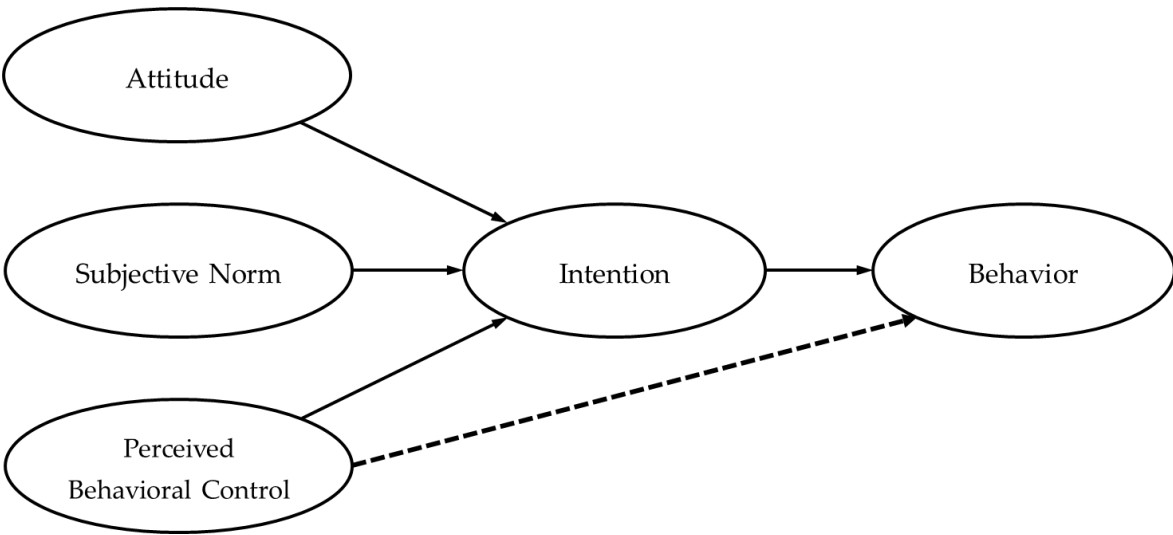

**Figure 1.** Theory of planned behavior. Adapted from Ajzen [29,30].

The TPB has been thoroughly tested in several behavioral domains. A few studies have also verified the model for mobility investments in households (e.g., [16,17,28]).

In general, the explanatory power of the TPB is well-proven, but some studies indicate room for improvement, mostly when it comes to normative aspects: within the TPB, subjective norms are understood to involve injunctive and descriptive elements. However, recent studies have shown that descriptive norms are a stronger behavioral predictor than injunctive ones [31,32]. Thus, it might be worthwhile to treat both aspects separately, or to consider descriptive norms only. In addition, some studies indicate that the TPB's predictive power could be further improved if it were extended by moral factors like personal ecological norms (e.g., [33–35]).

### 2.2. Norm-Activation Model and Value-Belief-Norm Theory

Some other action models instead focus on moral considerations as the main predictors of environmentally-relevant behaviors. Schwartz [36] suggested the Norm-activation model (NAM) to explain prosocial behavior; the model is frequently used in the environmental context. According to the NAM, pro-environmental behavior is driven by personal ecological norms (PN), i.e., feelings of moral obligation to act in an ecological way (Figure 2). The PN only lead to pro-environmental behavior if they are activated in a given situation. There are several slightly different versions of the model, which include different factors that lead to PN activation (see [37] for an overview). Most versions consider the awareness of need (AN), awareness of consequences (AC), ascription of responsibility (AR), perceived behavior control (PBC), and social norms as predictors of PN activation and formation.

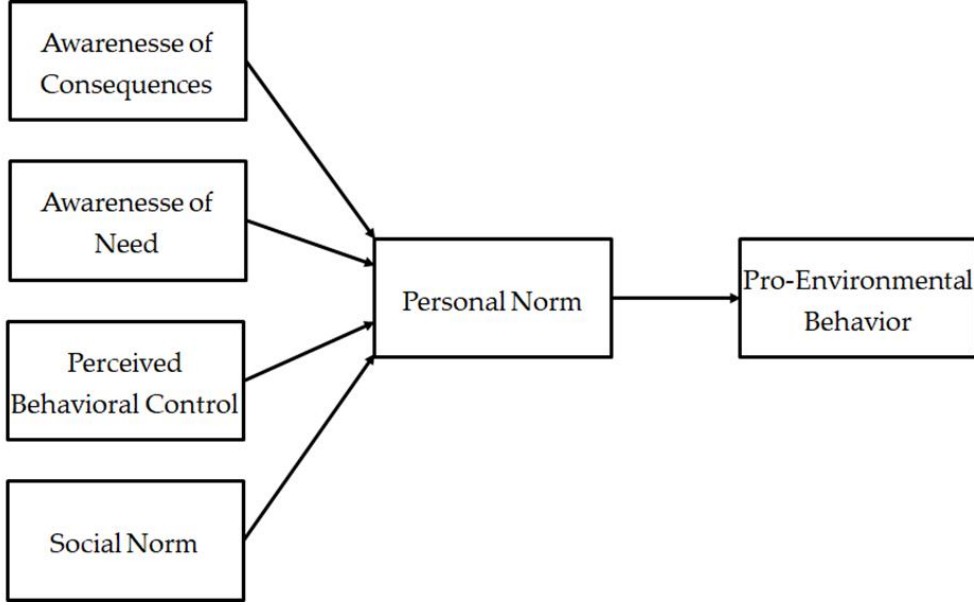

**Figure 2.** Norm-activation model (modified version, adapted from Klöckner [37]).

AN is a person's awareness that an environmental problem exists that needs to be alleviated; AC is one's perception that one contributes to the problem with one's own behavior or not; AR is the perceived personal responsibility of contributing to the problem solution; and the PBC is the perceived ability to make such a contribution. Social norms are commonly understood to involve descriptive and injunctive aspects—just as the subjective norms in the TPB. Thus, both concepts are widely used interchangeably.

Stern [38] suggested the value-belief-norm theory (VBN), another model focusing on the moral determinants of environmentally-relevant behaviors. The VBN has several similarities with the NAM. The central construct of the VBN is also the PN (Figure 3), as it is the direct predictor of environmentally-relevant behavior, and it first has to be activated in a given situation. However, in the VBN, norm activation is described to happen as a result of a cascade of several factors gaining relevance one after another. The first step in the VBN cascade is the formation of an ecological worldview (mostly measured using the new environmental paradigm; NEP). This formation depends on certain basic values: altruistic (i.e., caring for society) and biospheric values (i.e., caring for the natural environment) may contribute to one's ecological worldview, while egoistic values (i.e., seeking for personal benefits) may reduce it. Holding a strong ecological worldview then supports the perception of environmental problems (awareness of consequences; AC) and the ascription of one's responsibility to alleviate them (AR). The ecological worldview beliefs, AR and AC finally lead to PN activation and behavior, in so far as that activation is strong enough.

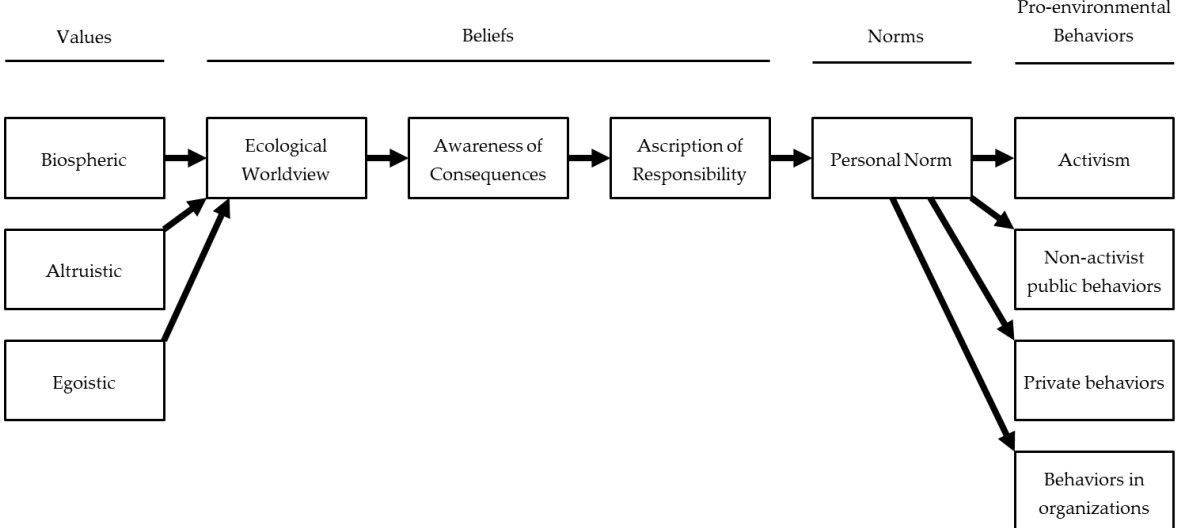

**Figure 3.** Value-belief-norm theory (adapted from Stern [38]).

The VBN explicitly covers different kinds of pro-environmental behaviors covering private sphere behaviors and behaviors in organizations. However, it should be noted that organizational behavior refers to staff behavior, rather than to decisions on the management level, even though the latter is not ruled out explicitly.

NAM and VBN were both also tested for mobility behaviors. As for the TPB, most of these investigations focused on everyday behavior, and there are only a few studies targeting mobility investments in households (e.g., [14,15]). Among those studies, the explanatory power of NAM and VBN was proven; however, the findings show that both models could be further improved by adding other 'non-moral' behavioral predictors (e.g., attitudes, see Sections 2.1 and 2.3).

### 2.3. Integrative Action Models

As we stated before, rational and moral-centered models have both proven to be of some value in explaining environmentally-relevant behavior and mobility behavior in particular. Their explanatory power could be further improved if both approaches were combined. Several integrative approaches have been suggested involving elements of rational and moral-centered models (e.g., [25,33–35,39,40]). Most of these analyses involved comparisons between the basic models (e.g., TPB, NAM, VBN) and integrated approaches, in which the explanatory power of the latter were generally higher. However, the number of studies using integrative models to explain mobility investment decisions is quite limited, and those that are available focus on household decisions only (e.g., [13,18,19]). We are not aware of any analyses testing integrative action models for mobility investments in organizations.

Most integrated models are based on the TPB. Intentions or similar constructs are treated as the main predictors of behavior, which are affected by other predictors. A most basic approach was suggested by Haarland and colleagues [33]: here, the TPB could be improved by adding PN as a further predictor of intentions (Figure 4).

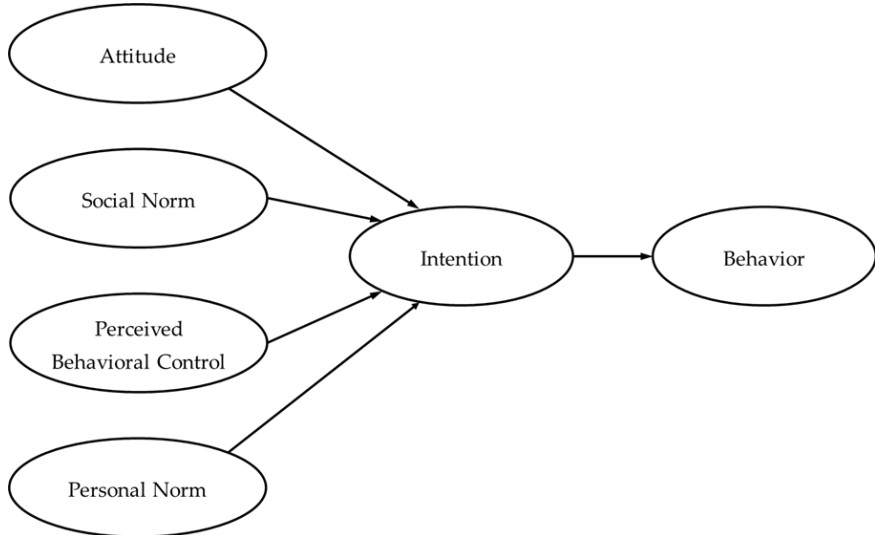

**Figure 4.** Integrative model suggested by Haarland et al. [33].

Other integrative models suggested more complex structures covering further constructs predicting attitudes, norms or PBC (e.g., [25,40]). In addition, subjective norms are sometimes not expected to have a direct effect on intentions but to be mediated by PN (e.g., [25,39]). Because none of the integrative models have yet been proven to be superior to the others, we stick to the simplest one suggested by Haarland and colleagues [33] in what follows.

### 2.4. The Relevance of Different Types of Social Norms

One shortcoming of most of the basic and integrative action models is that they treat social norms in terms of subjective norms as they are defined in the TPB. According to the TPB, subjective norms involve injunctive (i.e., the perceived social pressure exerted by significant others) and descriptive elements (i.e., the perceived behaviors of significant others). The empirical findings, however, suggest that injunctive and descriptive norms have a different influence on intentions and behavior. Nolan, Schultz, Cialdini, Goldstein and Griskevicius [31] found that descriptive norms were more relevant than injunctive norms for households' energy conservation intentions and their actual energy consumption. The same pattern was found for photovoltaic (PV) investments in private households [32].

In the field of innovative technologies (e.g., e-cars or PV investments), the relevance of descriptive norms can partly be explained by observability effects: if people notice that others use an innovative technology successfully, they become more likely to adopt the innovation themselves [41]. In addition, the relevance of descriptive and injunctive norms may be affected by measurement issues. When injunctive norms are measured, people are asked directly how much they were affected by social pressure. For the most part, people underestimate such influence, or they do not admit to complying to it (e.g., [42]). Descriptive norms assess social pressure in a way that is more indirect and may be less vulnerable to measurement issues. Thus, it may be worthwhile to treat both injunctive and descriptive norms separately, instead of integrating them into subjective norms.

### 2.5. Research Agenda and Questions

As we stated before, the goal of the study at hand is to analyze whether mobility investment in households and organizations are based on similar or different factors. The results could be used to improve policy strategies in this field, e.g., by tailoring them to the different target groups. Action models were used to compare decision making in both groups, as they allow us to identify which factors are more important than others. For this, we used the integrative action model suggested by Haarland and colleagues [33]. This approach involved the main factors of several action models—namely TPB, NAM

and VBN—that have already been proven to be of some value in explaining mobility investments, at least in households. Additionally, slightly different model variants were calculated, including different kinds of social norms, i.e., injunctive and descriptive norms, as their predictive power was found to be different in former studies.

For households, all of the constructs within the integrative model were expected to be of some relevance. As indicated by some recent studies, personal and social norms might be slightly more important than attitudes and PBC [18,34]. It is rather difficult to hypothesise such patterns for organizations, given the very small number of—mostly qualitative—studies on mobility investments in this field. The available studies, however, indicate what one would expect: organizations prioritize economic factors that grant them operational readiness. They tend to make investments if they can be expected to pay off, and if they are expected to fit the organizations' needs and possibilities (e.g., [21–23]). These findings are also supported by studies investigating other types of energy-related investments in organizations which are directly related to the production process (e.g., [43–48]). In action models, such factors are covered by non-normative factors, such as attitudes or PBC. This leads to the following hypotheses:

**Hypothesis 1.** *Mobility investments in organizations and households are driven by different factors.*

**Hypothesis 1a.** *Mobility investments in households are based on personal and social norms, rather than on attitudes and PBC.*

**Hypothesis 1b.** *Mobility investments in organizations are based on attitudes and PBC, rather than on personal and social norms.*

Regarding social norms, former research on energy-relevant investments suggests that descriptive norms are a better predictor than injunctive norms. In our study, we expect such pattern in both target groups:

**Hypothesis 2.** *Mobility investments are better explained by descriptive norms than by injunctive norms.*

### 3. Materials and Methods

We drew on data from two online surveys carried out in Germany in 2016 (study I) and 2018 (study II). We focused on investments in battery electric cars as a suitable type of alternative fuel vehicle for our study. Battery electric cars are innovative not only regarding the underlying technology but also regarding usage implications (e.g., dealing with limited range and a different charging technology). Furthermore, their economic and technical potentials have been demonstrated in previous studies, both for households (e.g., [49,50]) and organizations (e.g., [51]) in Germany. Study I assessed e-car investment decisions in households; study II targeted organizations. Both studies investigated investment decisions concerning new vehicles only, because there was no significant market for pre-owned e-cars at the time of our study (see Section 1).

#### 3.1. Samples

Both samples were recruited by an external panel provider. The participants to whom the decision problem applied were selected. For study I, we selected households planning a car purchase within the next two years. We only included household members that were involved in car investment decisions beforehand.

For study II, we focused on small organizations (up to 50 employees) running commercial fleets. We chose small organizations for two reasons: for one thing, most organizations (97.1%) in Germany are small (e.g., [52]). This is also true for many other countries. Thus, our findings might be relevant for several other organizations in many countries. For another thing, small organizations tend to have fewer formal rules when it comes to infrequent investments such as car purchases [53,54]. These organizations are run in a rather autocratic sense, in which single individuals or small groups make decisions occasionally. Thus, the decision situations in these organizations might be more similar to those

in households. In both the household and organization groups, all of the other sample characteristics were random.

At first, we investigated which business models generally met the capabilities of modern e-cars (i.e., businesses with several short distance tours). We then chose private care services for an example. Again, we only questioned people that were involved in the investment process (e.g., company owners, fleet managers). Several further screening questions were included (e.g., location, structure) in order to ensure the organizations' comparability. The fact that companies usually buy new cars was consistent with our data: 76% of the sample stated that they buy new cars, and only the remaining 24% stated that they buy new and pre-owned cars.

The samples from both studies differed in a number of demographic factors (Table 1). In study I, the sample consisted of more men (65.0%) than women (35.0%), whereas the ratio was 1:1 in study II. The level of educational attainment was higher in the sample of study I, as 38.7% of the participants held a university degree, whereas this related only to 14.9% of the participants in study II. Table 1 shows the demographics of both samples and the German average. The sample of study I differed from the general population in Germany according to gender and the highest educational attainment. The sample of study II differed from the general population in Germany according to the income per household and the highest educational attainment. We ascribe those differences to the criteria that were used during the screening of the appropriate participants.

**Table 1.** Overview of the social demographics in both samples.

| | Study I: Household Members (N = 227) | Study II: Commercial Fleet Owners (N = 101) | General Population in Germany |
|---|---|---|---|
| Age | | | |
| M | 45.6 | 45.9 | 44.4 |
| SD | 11.9 | 8.61 | N/A [1] |
| | | | |
| Gender | | | |
| Female | 35.0% | 50.0% | 50.7% |
| Male | 65.0% | 50.0% | 49.4% |
| | | | |
| Highest educational attainment | | | |
| No graduation | 0.0% | 0.0% | 4.0% |
| Secondary school | 9.8% | 3.0% | 29.6% |
| Intermediate school | 28.4% | 33.7% | 23.3% |
| Vocational baccalaureate and A level | 22.7% | 46.5% | 32.5% |
| University | 38.7% | 14.9% | 17.6% |
| Other graduation | 0.4% | 2.0% | N/A [1] |

Note: [1] The statistics were not available for the general population in Germany.

### 3.2. Measurements

The data were collected in two online surveys that were part of larger studies. Both online surveys also included choice experiments concerning e-car investments. The results of these experiments are reported elsewhere [55,56]. The analysis at hand will focus on the differences between households and organizations. In both studies, we measured all of the constructs of our integrative model using a 5-point Likert scale (1 = "I do not agree"; 5 = "I fully agree"). The scale values of all of the constructs consisting of more than one item were calculated as means of all of the items. The means and reliabilities of the scales are shown in Table 2. The different constructs were measured with one to three items. For three item measurements, Cronbach's α will be reported; for two item measurements, correlations will be reported.

**Table 2.** Overview of all of the variables and reliabilities.

|  | Study I: Household Members (N = 227) | Study II: Commercial Fleet Owners (N = 101) |
|---|---|---|
| Investment Intention |  |  |
| M | 2.62 | 2.62 |
| SD | 1.25 | 1.25 |
| Cronbach's α/Pearson r | N/A | r = 0.89 |
| Attitude |  |  |
| M | 3.51 | 3.28 |
| SD | 1.04 | 1.10 |
| Cronbach's α/Pearson r | α = 0.85 | N/A |
| Perceived behavioral control |  |  |
| M | 2.64 | 3.13 |
| SD | 1.04 | 1.03 |
| Cronbach's α/Pearson r | r = 0.28 | N/A |
| Personal ecological norm |  |  |
| M | 2.91 | 2.30 |
| SD | 1.08 | 1.01 |
| Cronbach's α/Pearson r | α = 0.83 | r = 0.77 |
| Social norms |  |  |
| Injunctive norm: General |  |  |
| M | 2.46 | 2.05 |
| SD | 1.17 | 0.99 |
| Cronbach's α/Pearson r | α = 0.85 | α = 0.89 |
| Descriptive norm: General |  |  |
| M | 2.25 | 1.38 |
| SD | 1.08 | 0.87 |
| Cronbach's α/Pearson r | r = 0.47 | N/A |
| Injunctive norm: Staff |  |  |
| M | N/A | 2.79 |
| SD | N/A | 1.17 |
| Cronbach's α/Pearson r | N/A | N/A |
| Descriptive norm: Competitors |  |  |
| M | N/A | 1.43 |
| SD | N/A | 0.56 |
| Cronbach's α/Pearson r | N/A | N/A |

Notes: Cronbach's α was calculated for all of the constructs measured with three items. Pearson's correlational coefficient r was calculated for all of the constructs measured with two items. All of the correlations were significant at $p < 0.001$. All of the other constructs were measured with one item only. The internal consistencies cannot be calculated for single-items measurement. The social norms referring to staff and competitors were only measured in the organization sample.

The investment intention was the dependent variable in both studies. It was measured with one item in study I ("I plan to purchase an electric car as the next car instead of a combustion engine car") and two items in study II ("I will support the purchase of electric cars in my organization in the future" and "I plan to purchase (also) electric cars for my organization"). The answering scale was a 5-point Likert scale (1 = "I do not agree"; 5 = "I fully agree") in both studies. In study II, the dependent variable was calculated as the mean of the two items. The correlation between those two items in study II was relatively high: r = 0.89; $p < 0.001$.

In order to measure the attitude of the decision behavior in study I, we used three items regarding the advantages of electric cars and their importance for the present and future mobility system ("Electric cars have many advantages compared to combustion engine cars"; "Electric cars are important means of transportation for private households in the future."; "Electric cars should play an important role in our mobility system."). The scale value was calculated as the mean values of those three items. The internal consistency was good, with Cronbach's α = 0.85. In study II, the participants rated their attitude towards the behavior on one item according to the statement "I think the purchase

of electric cars instead of cars with combustion engine for my organization is . . . ″ on a 5-point Likert scale (1 = "very bad"; 5 = "very good").

PBC was measured with two items in study I ("Currently, it would be difficult for me to purchase an electric car"; "My current circumstances of life determine whether I buy an electric car or a car with a combustion engine"). The two items correlated with r = 0.28, $p < 0.001$. The scale value was calculated as the mean values of both items. Therefore, the negative poled item ("Currently, it would be difficult for me to purchase an electric car") was recoded before. In study II, PBC was measured with one item ("It is possible for my organization to buy electric cars instead of cars with combustion engine").

The personal ecological norm was measured with a three item scale in study I ("Due to values important to me, I feel obliged to choose an electric car when purchasing a new car."; "Due to reasons of environment protection I will have a bad conscience if I purchase a car that is no electric car."; "No matter what other people do, my own principles tell me that it is right to purchase an electric car for reasons of environmental protection.") The scale value for the personal ecological norm was calculated as the mean values of those three items. The internal consistency was good, with Cronbach's $\alpha = 0.83$. Two items were used in study II ("Due to values important to me, I feel obliged to purchase an electric car for my organization."; "Due to reasons of environmental protection I will have a bad conscience if I purchase a combustion engine car for my organization instead of an electric car.") and correlated with r = 0.77, $p < 0.001$. The scale value was calculated as the mean values of both items.

We measured several types of social norms in order to compare their decision relevance for e-car investments. In study I, we measured the injunctive and descriptive norms as suggested by Ajzen [30]. For the injunctive norms, three items were used ("People who are important to me think that I should purchase an electric car."; "People who are important to me, will support me if I purchase an electric car."; "People who are important to me signify that I should consider electric cars when purchasing a car."). The scale value of the injunctive norms was calculated as the mean values of the three items. The descriptive norms were measured with two items ("Electric cars a currently easy to observe in their daily usage."; "People in my private setting are currently driving an electric car."). The scale value of the descriptive norms was calculated as the mean values of both items. Our scale reliability analyses suggested that we should treat injunctive and descriptive norms separately, rather than combining them into a general subjective norm. Cronbach's $\alpha$ was high for the injunctive norm items ($\alpha = 0.85$) and could not be improved by adding the descriptive norm items. The two remaining items referring to descriptive norms correlated with r = 0.47, $p < 0.001$.

For the organizations (study II), a larger number of social norms were measured, as we expected that several social groups could affect the decision process. In general, e.g., in the household context, the social norms were measured for private individuals referring to significant others, such as family, friends, neighbors, or similar. Such rather unspecified groups may also be important for decision makers' organizations. Thus, their influence was measured using three items referring to injunctive norms ("People who are important to me think that I should purchase an electric car instead of a combustion engine car for my organization."; "People who are important to me support me if I purchase an electric car instead of a combustion engine car for my organization."; "People who are important to me signify that I should consider electric cars when purchasing a car for my organization."). The scale value was calculated as the mean values of those three items. The descriptive norm was measured with one item ("People in my private setting are currently driving an electric car."). Again, our analyses revealed a high reliability for the injunctive norm scale (Cronbach's $\alpha = 0.89$), suggesting the separation of injunctive and descriptive norms.

However, there may be further relevant social groups within the organizational context. First, people in the same situation as the decision makers may set relevant examples, mainly for other decision makers in organization. In addition, colleagues or customers might favor certain behaviors. We measured these influences as well by asking for the perceived social

norms set by other firms and staff. Here, we found it inappropriate to measure injunctive and descriptive norms for colleagues and customers. In our view, injunctive norms cannot be measured for other firms, as they would not explain to their competitors what they expected from them. Thus, we relied on descriptive norms by asking what the decision makers perceive their competitors to do. We used one item to measure the descriptive norm ("Many other care services are currently driving electric cars."). For staff members, we measured only injunctive norms with one item ("I think that the staff of my organization would support that I purchase an electric car instead of a combustion engine car."), as we found it unlikely that the decision makers knew whether their employees had electric cars or not (i.e., a descriptive norm). We decided not to consider social norms covering customer expectations, as we found it most unlikely that decision makers would know their expectations and current behaviors.

## 4. Results and Discussion

In this section we will first present our results for both target groups' households and organizations separately, and discuss the hypotheses' validation. For each group, several models were calculated, including several predictors and different types of social norms. We will discuss the relevance of each predictor included (see Hypotheses 1, Section 2.5), and the differences in the overall of the explained variances of the models depending on different social norms (Hypothesis 2). In the second part, we will discuss the differences between both target groups, focusing on the models with the highest explained variance for each group.

### 4.1. Households

For households, Hypotheses 1a and 2 were confirmed. Two linear regression models were calculated for e-car investment decisions in households (see Table 3). Both models included different types of social norms referring to significant others in general (i.e., "people who are important to me"; see Section 3.2). Model 1 involved injunctive norms, and model 2 involved descriptive norms.

**Table 3.** Regression results for e-car investment intentions in the household sample.

|  | Model 1 | Model 2 |
|---|---|---|
| Attitudes | **0.11** | **0.18** |
|  | (0.04) | (0.00) |
| Perceived Behavioral Control | 0.07 | **0.09** |
|  | (0.12) | (0.04) |
| Personal Norms | **0.46** | **0.50** |
|  | (<0.001) | (<0.001) |
| Social Norms Injunctive Norms |  |  |
|  | **0.32** |  |
| Descriptive Norms | (<0.001) | **0.32** |
|  |  | (<0.001) |
| Adjusted $r^2$ | 0.61 | 0.64 |

Notes: The upper figures in each cell are standardized $\beta$ coefficients. The *p*-values are displayed in brackets. The statistically significant results are printed in bold.

As expected, personal and social norms were found to be highly relevant predictors of investment intentions across both models. Personal norms were the strongest predictor (model 1: $\beta = 0.46$, $p < 0.001$; model 2: $\beta = 0.50$, $p < 0.001$), followed by social norms ($\beta = 0.32$, $p < 0.001$ in both models). Attitudes also had a significant, but rather weak, influence on investment intentions in both models (model 1: $\beta = 0.11$, $p < 0.05$; model 2: $\beta = 0.18$, $p < 0.05$). PBC only had a significant, but rather weak, influence in model 2 ($\beta = 0.09$, $p < 0.05$), but not in model 1 ($\beta = 0.07$, n.s.).

As expected, the analyses showed that more variance ($r^2 = 0.64$) was explained if descriptive norms were integrated, compared to injunctive norms ($r^2 = 0.61$). All of the

$r^2$ values are adjusted determination coefficients. These results confirm hypothesis 2 for households, even though the difference was rather small.

### 4.2. Organizations

For organizations, four linear regression models were calculated, including different types of social norms (Table 4). As for the households, models 1 and 2 involved injunctive and descriptive norms referring to unspecified significant others. Model 3 involved injunctive norms referring to staff members' expectations, and model 4 involved descriptive norms referring to the perceived behavior of competitors (see Section 3.2).

**Table 4.** Regression results for e-car investment intentions in the organization sample.

|  | Model 1 | Model 2 | Model 3 | Model 4 |
|---|---|---|---|---|
| ATT | **0.51** | **0.55** | **0.36** | **0.54** |
|  | (<0.001) | (<0.001) | (<0.001) | (<0.001) |
| PBC | **0.21** | **0.19** | **0.23** | **0.19** |
|  | (0.00) | (0.00) | (<0.001) | (0.00) |
| PN | **0.19** | **0.24** | **0.16** | **0.26** |
|  | (0.03) | (0.00) | (0.04) | (0.00) |
| SN |  |  |  |  |
| INJ: General | **0.16** |  |  |  |
|  | (0.04) |  |  |  |
| DES: General |  | 0.10 |  |  |
|  |  | (0.09) |  |  |
| INJ: Staff |  |  | **0.36** |  |
|  |  |  | (<0.001) |  |
| DES: Competitors | **0.51** | **0.55** | **0.36** | **0.54** |
|  | (<0.001) | (<0.001) | (<0.001) | (<0.001) |
| Adjusted $r^2$ | 0.74 | 0.74 | 0.79 | 0.72 |

Notes: The upper figures in each cell are standardized β coefficients. The *p*-values are displayed in brackets. The statistically significant results are printed in bold. ATT = Attitudes; PBC = Perceived Behavioral Control; PN = Personal Norms; SN = Social Norms; INJ = Injunctive Norms; DES = Descriptive Norms.

For organizations, Hypothesis 1b could only partly be confirmed. As expected, attitudes had a rather strong significant influence on the investment intentions across all of the models, ranging between β = 0.36 and β = 0.55. Thus, attitudes were the strongest predictor in models 1 (β = 0.51, *p* < 0.001), 2 (β = 0.55, *p* < 0.001) and 4 (β = 0.54, *p* < 0.001). In model 3, attitudes had the same influence as social norms (i.e., staff-related injunctive norms; attitudes: β = 0.36, *p* < 0.001; social norms: β = 0.36, *p* < 0.001). Contrary to our expectations, PBC had a rather low, but significant, influence across all of the four models, ranging between β = 0.16 and β = 0.23. Only in model 1 (β = 0.21, *p* < 0.01) was it a stronger predictor than personal (β = 0.19, *p* < 0.05) and social norms (β = 0.16, *p* < 0.05). In models 2 and 4, personal norms (model 2: β = 0.24, *p* < 0.01; model 4: β = 0.26, *p* < 0.01) were more relevant than PBC (model 2: β = 0.19, *p* < 0.01; model 4: β = 0.19, *p* < 0.01). In model 3, social norms (β = 0.36, *p* < 0.001) were more relevant than PBC (β = 0.16, *p* < 0.01). Thus, in model 3, social norms were as relevant as attitudes.

Hypothesis 2 needed to be rejected for organizations. Across all of the four models, a rather large share of the investment intention's variance could be explained ($r^2$ = 0.72 to 0.79). Most unexpectedly, the models including descriptive norms were not found to be superior to those including injunctive norms. In fact, model 3 (referring to staff-related injunctive norms) had the strongest predictive power ($r^2$ = 0.79), followed by models 1 and 2, which were equally strong ($r^2$ = 0.74), no matter if they involved injunctive or descriptive norms of unspecified significant others. Model 4 (referring to competitors' descriptive norms) was the weakest of all of the four models ($r^2$ = 0.72).

The influence of social norms strongly depended on the norms' type. On the one hand, the descriptive norms in models 2 and 4 were found to be insignificant, no matter if they referred to significant others in general (β = 0.10, n.s.) or to competitors (β = 0.01, n.s.). On

the other hand, both injunctive norms were found to be significant. As for those referring to significant others in general (model 1), they were the weakest of all of the predictors ($\beta = 0.16$, $p < 0.05$); as for those referring to staff expectations in particular (model 3), they were as relevant as attitudes ($\beta = 0.36$, $p < 0.001$).

### 4.3. Difference between Households and Organizations

The final models for households and organizations, i.e., those with the highest degree of explained investment intention variance (i.e., adjusted $r^2$ values), are presented in Figures 5 and 6. As social norms, unspecified descriptive norms were included for households, and staff-related norms were included for organizations. The models we chose for the further analyses not only involved the highest explained variances in both target groups but also highly significant $\beta$ coefficients for all of the predictors. This allows a better understanding of the interplay of these predictors within each model.

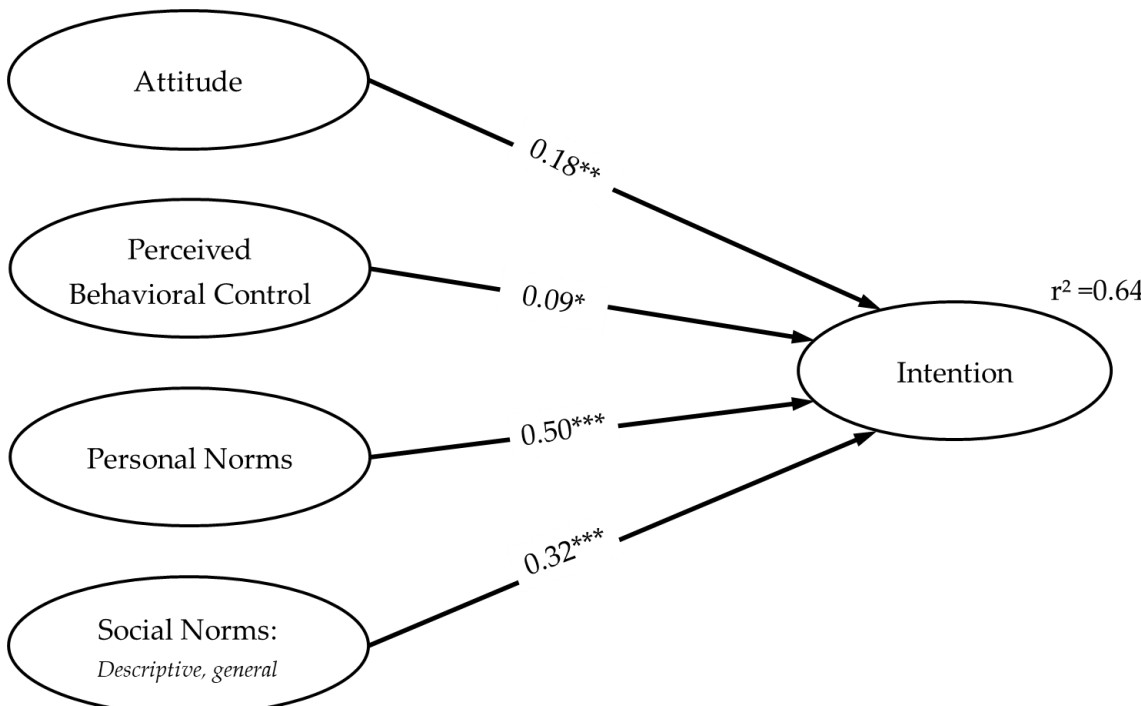

**Figure 5.** Final model for e-car investments in households. Notes: * $p < 0.05$; ** $p < 0.01$; *** $p < 0.001$.

Generally, we found our integrative model to be more suitable for the explanation of investment intentions in organizations than in households. The final model for organizations explained 79% of the intention variance, compared to 64% in the household sector. In fact, all of the models that were calculated for organizations had higher r squares than those that were calculated for households (see Tables 3 and 4). One goal of our study was to investigate whether common action models that have been tested mainly for households are also suitable for investment decisions in organizations. The results indicate such transferability, at least when it comes to e-car purchases. Given the higher explained variances in organizations, it may be that professional decision making is even more elaborated than private-sphere behavior.

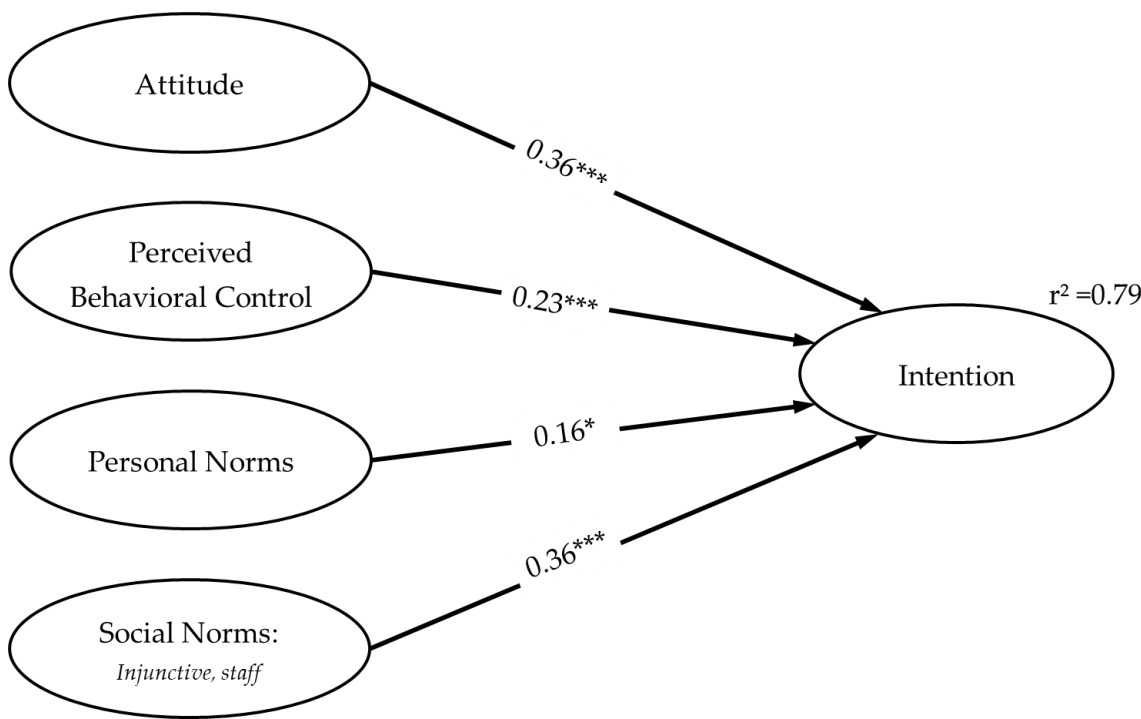

**Figure 6.** Final model for e-car investments in organizations. Notes: * $p < 0.05$; *** $p < 0.001$.

First of all, our findings show that all of the predictors of our integrative model had some relevance in both target groups. As we had expected, the importance of each single predictor differed considerably between the groups. However, these differences did not always show the pattern we assumed: we can confirm our idea that households mainly base their investment decisions on normative factors, but we need to reject the assumption that organizations did the opposite. Here, our findings merely confirmed that attitudes were highly relevant, while PBC was of medium strength only. Additionally, normative factors—namely certain social norms—were found to be more important than expected. They were as relevant as attitudes for mobility investment intentions.

It should be noted that the most relevant social norms in organizations were not the same ones as for households. In organizations, social norms referring to injunctive staff behaviors were particularly relevant; for households, descriptive norms referring to significant others were most important. Only the latter findings support the previous research suggesting that descriptive norms have a higher predictive power than injunctive ones (see Section 2.4). These findings could not at be replicated for organizations, in which injunctive norms always had a stronger influence than descriptive ones. In all probability, people in organizations consider the perceived expectations of others more strongly in the decision-making process. Staff members may be a key group, as they are the ones using the goods (e.g., the cars) that are purchased.

## 5. Limitations

The study involves some room for improvements which should be considered in future analyses. We questioned two different participant groups considering mobility investments in either households or organizations. The analyses investigating the differences between the household and organizational decisions would be more reliable if the same people were asked in both settings, i.e., what factors drive their private vs. organizational car purchases. Such studies would be worthwhile, but very hard to conduct. It is complicated to gain a sizeable number of participants from comparable organizations involved in car investment processes. If that group were further restricted to people who currently consider private car purchases, it would be somewhat impossible to find a sufficient number of participants.

It may be an option to ask the people in charge of the organization's fleet what drove their latest private car purchases, no matter how long ago. We found that approach unsuitable, though: the e-car market has changed rapidly between the last few years, and so have the decision situations. In addition, such retrospective questions are vulnerable to recall biases (e.g., [57]).

Across both questionnaires, we did not always use the same number of items to measure all of the constructs of our models. For each construct, the number of items ranged between one and three. These restrictions had to be made, as our study was part of a greater study that did not offer sufficient room for more items. In future studies, our data should be replicated using at least three items per construct.

The household data were collected in 2016. One might say that household attitudes towards e-mobility purchases have changed fundamentally since then, given the dynamic developments in this field. More longitudinal research is needed to clarify whether such changes have really happened. So far, the limited number of available review studies do not suggest that there have been fundamental changes in attitudes over time [11,58]. This is also true for other types of energy-relevant household investments. Data from several decades indicate that the main investment drivers are rather stable [59].

In the organizational sample, social norms were measured in three ways: we assessed descriptive and injunctive norms for (unspecified) significant others; for competitors, only the descriptive norms were measured, and for staff members, only the injunctive norms were measured. We made these restrictions as we found it unrealistic that the decision makers were aware of injunctive norms set by competitors and the actual behavior of staff members (i.e., what kind of car they drive; see Section 3.2). That assumption could be further examined in future studies. In all probability, the other two types of social norms (i.e., injunctive norms for competitors and descriptive norms for staff) could be involved, including an answer option saying "I do not know/I cannot tell".

Our analyses only involved one certain type of small organization, namely private care services. We chose this context because small organizations are the most common form in several countries (I); they have decision structures, which are somewhat comparable to those in households (II); and private care services have mobility demands which fit the capabilities of modern e-cars (III). However, there is a need for further analyses to investigate the decision situations in other sectors and larger organizations.

## 6. Conclusions and Policy Implications

Our analyses show that action models have some value in explaining e-car investment decisions in both households and organizations. A great deal of variance can be explained by the integrated model we chose, no matter in which context. As one would expect, the relevance of certain predictors differed across both target groups. Decision makers in households rely on personal ecological norms and social norms in terms of descriptive norms of significant others' behavior. Decision makers in organizations base their decisions on attitudes and social norms reflecting staff members' expectations. Thus, we could only partly prove our assumptions: as expected, households mainly rely on normative factors, but organizations also consider them strongly.

Still, it is most worthwhile, in our view, to use the same action models for both target groups. For one thing, they have considerable explanatory power; for another thing, they might help to further investigate the differences between households and organizations. Future analyses could also be expanded to target group comparisons focusing on decision objects other than e-car investments. Such analyses—and replications of our study—might help us to gain further insight into the nature of these differences. Target group-specific model modifications could be carried out based on the results of those investigations, if necessary.

The findings of our study also come with some important implications for policy makers. If the decision makers in households and organizations base their e-car investments on different factors, group-specific policy measures might be necessary. Such measures

should be tailored to each target group, focusing on the most relevant decision factors. For households, such measures should clearly involve information strategies targeting personal ecological norms. Different media could be used here, underlining the ecological benefits of e-cars—especially compared to conventional ones (for more information on ecological benefits, see e.g., [60,61]). Descriptive norms should rather be targeted with campaigns involving social models already using e-cars. They could tell others about their (positive) experiences, or possibly even offer test drives and thus help to reduce reservations. Implementing such approaches is naturally quite challenging, but also rather effective.

Policy measures for organizations should, for one thing, target general attitudes concerning e-car investment. Certain information strategies could be used here, showing the general benefits of e-cars, e.g., from an economical point of view. The information could, for instance, clarify that e-cars are often economically competitive with conventional cars, if their purchasers not only look at the investment price but the total life cycle cost. For this, online calculators could be provided (or further promoted), in which investors could make calculations for their own driving profiles (see e.g., [62]) for one calculator example showing the competitiveness of different conventional and e-cars). Such websites could also involve further calculators allowing organizations to check whether current e-cars match their driving profiles.

Our analyses show that staff members also have a large influence on the investments. If decision makers feel that the staff would favor sustainable investment—such as e-car purchases—they are more likely to make such investments. Certain strategies supporting staff members' sustainability expectations may be worthwhile. For this, staff representatives should be targeted, rather than decision makers themselves. Certain information could be used here, elaborating the benefits of e-cars from an economic point of view. Among others, collaborations with labor unions may be worthwhile, especially as several of them already support sustainability goals.

Altogether, our analysis shows that there is still a fundamental need for research on mobility investment decisions. On the one hand, there are some positive tendencies, as an increasing number of studies investigate such investments in households. This work must continue to capture the dynamic development in this field. On the other hand, there is a research gap regarding mobility investments in organizations, even though they are the more important target group, as they buy more new cars. Thus, little is known about the main drivers of mobility investments in organizations. This lack of knowledge might also lead to suboptimal policy strategies. Gaining more insights into the nature of these decision processes and the differences between the target groups might help to support sustainable investments more effectively. This may be an important step to accelerate the energy and mobility transition.

**Author Contributions:** Conceptualization, I.K.; methodology, S.B. and I.K.; formal analysis, I.K., A.B. and S.B.; investigation, S.B., A.B. and I.K..; writing—original draft preparation, I.K. and A.B.; writing—review and editing, E.M.; visualization, I.K.; project administration, E.M.; funding acquisition, E.M. and I.K. All authors have read and agreed to the published version of the manuscript.

**Funding:** This study was conducted as part of the KOPERNIKUS project ENavi (system integration) funded by the German Federal Ministry of Education and Research.

**Institutional Review Board Statement:** Ethical review and approval were waived for this study, as it did not concern ethical issues.

**Informed Consent Statement:** Informed consent was obtained from all subjects involved in the study.

**Data Availability Statement:** Data can be made available by request to the authors.

**Acknowledgments:** We would like to thank all of those who scientifically or practically supported our project. Special thanks go to our colleagues Annalena Becker and Charlotte Schmid from the Otto-von-Guericke-University Magdeburg, who gave us valuable input during the preparation of this paper.

**Conflicts of Interest:** The authors declare no conflict of interest.

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
