# Peer review of "Are Professionals Rationals? How Organizations and Households Make E-Car Investments"

_sustainability, doi:10.3390/su13052496_

Round 1
Reviewer 1 Report
Improvements have been made.
Author Response
Thank you for your review.
Reviewer 2 Report
Some comments regarding the statistical analysis envisaged: you are using mostly ordinal data (Lickert scales), but you use statistical techniques reserved for numerical variables (means, standard deviations, correlation coefficients, regression models). This is often seen in published articles, but nevertheless, this is not correct. With ordinal data it makes no sense doing arithmetical calculations. There are non-parametric techniques that should be applied in these cases, but never methods reserved exclusively to numerical scaled variables.
Independently of these considerations, you report adjusted determination coefficients, but you interpret them as they were determination coefficients (that can correctly represent the proportion explained by a regression model of the endogenous variable used; but this can not be done for the adjusted determination coefficients, as you state in lines 500 and following). Again a variance can be calculated on numerical variables, non on ordinal 5-point Lickert data.
Also, in table 3 you use the writing R2 (which is for the determination coefficient, non the adjusted usual notation), and later on you write r2.
Author Response
We already responded to the comment why we find it appropriate treating the data gathered with a 5-point Likert scale as metric. This is most common in Social Sciences (see e.g., Döring & Bortz, 2016; Rohrmann, 1978; https://pascal-francis.inist.fr/vibad/index.php?action=getRecordDetail&idt=12743193). The authors in this field point out that equidistant scale gradiations should be used. We took care for that be using identical and well-established gradiations for the Likert scales across both studies.
We also find it appropriate using the adjusted determination coefficient instead of the non-adjusted. In our view, the adjusted coefficient is even the better: As far as we know It also has explanatory power for the endogenous variable and it is not affected by the number of predictors included (like the non adjusted determination coefficient).
To clarifiy that we used the adjusted r² instead of the non adjusted, we added the term "adjusted" in the tables. In the text, we already clarified that all r² values presented are adjusted (at the very end of section 4.1).
Reviewer 3 Report
The current manuscript displays a significant number of improvements as compared to previous version.
The title has been changed and, I think, is more relevant and it even sounds better than previous.
The authors explain more clearly (lines 650-655) the relevance of the data for the investigated topic.
The newly included paragraph in the discussion topics about the novelty and the relevance of data in the current context is quite compelling and reveals using straightforward arguments the author’s point of view.
Moreover, the authors included in the methodology section the information on the method they use to aggregate data.
In the light of these comments, it is my strong belief that this version of manuscript meets the quality criteria to be published in this form.
Author Response
Thank you for your review.